# Magnesium Zirconate Titanate Thin Films Used as an NO_2_ Sensing Layer for Gas Sensor Applications Developed Using a Sol–Gel Method

**DOI:** 10.3390/s21082825

**Published:** 2021-04-16

**Authors:** Pei-Shan Huang, Ke-Jing Lee, Yeong-Her Wang

**Affiliations:** Institute of Microelectronics, Department of Electrical Engineering, National Cheng Kung University, No. 1, University Road, Tainan 701, Taiwan; ytr0386685@gmail.com (P.-S.H.); Luke.k.j.lee@gmail.com (K.-J.L.)

**Keywords:** gas sensor, magnesium zirconia titanate, thin film, sol–gel method, nitrogen dioxide

## Abstract

Magnesium zirconate titanate (MZT) thin films, used as a sensing layer on Al interdigitated electrodes prepared using a sol–gel spin-coating method, are demonstrated in this study. The p-type MZT/Al/SiO_2_/Si structure for sensing NO_2_ is also discussed. The results indicated that the best sensitivity of the gas sensor occurred when it was operating at a temperature ranging from 100 to 150 °C. The detection limit of the sensor was as low as 250 ppb. The sensitivity of the MZT thin film was 8.64% and 34.22% for 0.25 ppm and 5 ppm of NO_2_ gas molecules at a working temperature of 150 °C, respectively. The gas sensor also exhibited high repeatability and selectivity for NO_2_. The response values to 250, 500, 1000, 1500, 2000, 2500, and 5000 ppb NO2 at 150 °C were 8.64, 9.52, 12, 16.63, 20.3, 23, and 34.22%, respectively. Additionally, we observed a high sensing linearity in NO_2_ gas molecules. These results indicate that MZT-based materials have potential applications for use as gas sensors.

## 1. Introduction

The rapid development of technology and industry has resulted in rapid increases in the levels of pollution in the environment. Various types of hazardous gases (such as H_2_S, CO, NO_2_, NH_3_, and ethanol) are routinely released from industrial and agricultural processes on a daily basis or emitted as vehicle exhaust. These gases are dangerous for human health and the environment when their concentrations exceed a critical threshold limited value (TLV) [1,2]. Among these gases, NO_2_ is the most harmful air pollutant. In the presence of toxic, irritating gases, NO_2_ causes the degradation of lung tissue and lowers the immune system even at relatively low concentrations [3,4]. Presently, highly sensitive, real time detection of trace NO_2_ using inexpensive, convenient sensor devices is still urgently needed to protect public health. Metal oxide resistive-type NO_2_ sensors have attracted a significant amount of attention due to their high performance ability. Metal oxide exhibits excellent gas sensitive properties due to its high specific surface area and enhanced surface reactivity [5,6,7,8]. In addition, the high operating temperatures of some metal oxide-based sensors can lead to increases in power consumption and reductions in the lifetime of sensors [5,6,7,8,9]. Metal oxide resistive-type NO_2_ sensors are also simple and inexpensive to manufacture. Unfortunately, most metal oxide sensors have to work at elevated temperatures (200–400 °C), which leads to problems related to high energy consumption and a risk of gas explosions. Therefore, there is now a strong demand for developing reliable, accurate, and cost-effective gas sensors with enhanced sensitivity, selectivity, and response times to detect harmful gases.

Gas sensing characteristics have been demonstrated in a variety of materials, including binary metal oxides, carbon materials, and perovskite oxides. Among them, perovskite oxides have attracted a significant amount of attention owing to their simple fabrication, good stability, remarkable electron mobility, structural tenability, and chemical composition through partial substitution with aliovalent elements. Magnesium titanate (MTO), as a perovskite material, has good physical and electrical properties, including a moderate dielectric constant, low dielectric loss, and high temperature stability, and is commonly used in microwave dielectrics [10,11,12]. Zirconium dioxide (ZrO_2_) has three main polymorphs (monoclinic, tetragonal, and cubic), used for electrochemical gas sensing at high operating temperatures [13,14,15,16]. The oxygen ions in ZrO_2_ can be actively transported at high temperatures (400–700 °C), limiting its wide application. In addition, there are a few reports indicating that Y_2_O_3_-stabilized ZrO_2_ can be utilized for the sensing of a variety of gases and can enable operation at lower temperatures [17,18,19,20]. Bang et al. reported that the NO_2_ gas sensing capabilities of SnO_2_-ZrO_2_ NWs with a particular shell thickness are better than those of pristine SnO_2_ sensor in terms of sensor temperature and sensor response [21]. Owing to differences in work functions, electrons will flow from SnO_2_ to ZrO_2_. Myasoedova et al. developed a sol–gel method for SiO_2_/ZrO_2_ composite films [22]. The sensitivity of the sensor up to 1060 ppm high concentrations of NO_2_ was low at 25 °C, and the response was only 44%. Yan et al. used ZrO_2_-HS, ZrO_2_–S, and ZrO_2_-R hydrothermal and solvothermal methods to successfully synthesize a ZrO_2_-R sensor, which showed the highest response towards 30 ppm NO_2_ (423.8%) at room temperature and a quite high sensitivity of 198% for detecting 5 ppm NO_2_ [23]. Mohammadi et al. found a remarkable response towards low concentrations of NO_2_ gases at 150 °C [19].

Titanium and zirconium have a similar atom radius (Ti: 2 Å, Zr: 2.16 Å) and the same valence state (+4). As a result, it is possible for a Zr atom to be incorporated into an MTO lattice through substitution for a Ti atom. Because substituting Zr^4+^ for Ti^4+^ can create some structural defects, the recombination of electrons and holes can be suppressed due to their trapping effects [24]. Since more carriers can interact with gas molecules, adding Zr into MTO can enhance its sensing properties. Adding Zr can also alter the morphology of the structure, leading to a rougher surface. Because a rougher surface provides a larger specific surface area for adsorbing more gas molecules, a stronger response can be achieved [25]. Interdigitated geometry was adopted for the sensing electrodes used in this work. Interdigitated electrodes are widely utilized in technological applications, especially in the field of biological and chemical sensors, because they are inexpensive, easy to fabricate, and have high sensitivity [26].

In this work, the sensing material (magnesium zirconate titanate (MZT)) was prepared using the sol–gel method (because the sol–gel method requires considerably less equipment and is potentially less expensive than other alternatives). The gas sensing measurements showed that the MZT-based gas sensor could detect ppb-level NO_2_ at less than 3 ppm. The low-cost route and high sensing performance of MZT thin films make them promising candidates for NO_2_ gas sensor applications.

## 2. Materials and Methods

The MZT based gas sensor proposed in this work was fabricated on an Si/SiO_2_ substrate. First, a pattern of interdigitated electrodes was defined through photolithography. Second, Al interdigitated electrodes were deposited using RF sputtering. Third, a 0.5 M MZT solution was synthesized as follows: An appropriate amount of magnesium acetate was added to glacial acetic acid via stirring and then heated on a 120 °C hot plate to obtain solution A. Titanium isopropoxide was dissolved into 2-methoxythanol via stirring to produce solution B. Zirconium n-propoxide was added to the acetylacetone via stirring and then heated on a 120 °C hot plate until dissolution to generate solution C. Solutions A, B, and C were then mixed. Then, 0.5 M MZT solutions were prepared by adding 2-methoxythanol, as shown in Figure 1a. Finally, the MZT solution was spin-coated onto the Al electrodes and baked at 100 °C for 10 min. Figure 1b shows a schematic of the MZT/Al/SiO_2_/Si device and the measurement setup. The fabricated device was placed on the carrier in a closed stainless-steel chamber. The carrier could be heated to the desired working temperature so that the responses of a gas sensor operating at different temperatures could be measured. The Keithey 2400 source meter was used to offer a fixed voltage in order to measure either the current or the resistance of the sample. When the target gas was injected into or removed from the chamber, the variations in the current curve were observed using Labview.

## 3. Results

The electrode pattern on the mask was designed as interdigitated geometry. The finger widths and gaps were 10 μm and 5 μm, respectively, as shown in Figure 2a. An SEM image of the fabricated interdigitated electrodes is shown in Figure 2b. The images showing the surface morphology of the pure MTO and the doped MZT thin film are provided in Figure 2c,d, respectively. From the images, it can be seen that many nanoparticles are anchored on the surface of the MZT thin film upon doping with Zr. Therefore, the surface of the MZT thin film exhibits a rougher appearance.

The crystallinity studies were conducted using X-ray diffraction (XRD) (Germany/D2 Phaser) with Kα1 (λ = 0.15405 nm) radiation. The XRD patterns of the MTO and MZT thin films deposited on the glass substrates are shown in Figure 3a,b. The peaks around 25° are from the glass substrate, and no other apparent diffraction peaks existed. This indicates that the MTO and MZT thin films were amorphous. The atomic force microscopy (AFM) in Figure 3c,d revealed the surface morphology of the undoped MTO thin film and the MZT thin film, respectively. The roughness of the MTO and MZT thin films were 0.317 and 15.7 nm, respectively, where greater roughness enhanced the effective surface area, thus increasing the number of adsorption sites and thereby improving the gas response of the films [25]. 

The morphology of the samples was explored using transmission electron microscopy (TEM, JEOL JEM-2100F CS STEM). In addition, energy-dispersive X-ray spectroscopy (EDAX), coupled with TEM, was used to carry out an elemental analysis. The low-magnification TEM image shown in Figure 4a reveals the thickness of the MZT film to be approximately 62.6 nm. The thickness of the Al electrodes is 111.5 nm. Figure 4b shows the EDAX analyses taken from the MZT layer grown on an Al substrate. The result confirms the chemical purity of the sample with only Ti, oxygen, and Mg or Zr present in the spectra. A compositional analysis of the spectra revealed the Mg or Zr and Ti, demonstrating the typical 8.6%, 6.7%, or 18.6% stoichiometry of the MZT layer, respectively.

The X-ray photoelectron spectroscopy (XPS, PHI 5000 VersaProbe) analyses were performed on a Perkin–Elmer PHI 5000 Versaprobe system. The MZT thin film sample was used in the XPS measurement. The XPS survey spectra of the MZT thin film were obtained after 90 s of Ar^+^ ion sputtering, thereby representing the bulk layer of the MZT thin film. As shown in Figure 5a, the atomic percentages of Mg, Zr, Ti, and O were 5%, 6%, 17%, and 70%, respectively. The XPS signals corresponded to Mg 1s, Zr 3d, Ti 2p, and O 1s. The XPS spectra for the Mg 1s (Figure 5b) were located at 1303.7, which matched well with the Mg^2+^ [27]. The peaks at 183 eV and 185.27 eV were attributed to Zr 3d5/2 and Zr 3d3/2, respectively [25]. The binding energy for Ti 2p3/2 and Ti 2p1/2 can be observed at 458.6 and 464.5 eV, respectively, which were typical for Ti^4+^ [28]. The peaks for the O 1s core level could be consistently fitted using two different near-Gaussian subpeaks centered at 530.3 and 531.9 eV, which were related to the lattice oxygen and the oxygen vacancies, respectively [29].

The gas sensing properties of the sensor were measured at different temperatures, and the variations in the sensor current were monitored at an applied voltage of 3 V. The response of the sensor was defined as $\frac{\left(R_{a}-R_{g}\right)}{R_{a}}\times 100\left(\%\right).$ The response values of the MZT thin film gas sensor, with NO_2_ concentrations ranging from 250 ppb to 5 ppm at different operating temperatures, are shown in Figure 6. A temperature of 150 °C was chosen as the optimum operating temperature. Although the best response values were obtained at 100 °C, the response and recovery times were poorer than those at 150 °C. The response values of the gas sensor at different operating temperatures with various NO_2_ concentrations are shown in Figure 7.

The current variations in the sensor due to the different NO_2_ concentrations at 150 °C are illustrated in Figure 8. Based on the trend in the current curves, the sensing material based on the MZT thin film exhibited the sensing characteristics of p-type materials. The response was observed to increase with higher NO_2_ concentrations. At lower concentration levels, the test gas molecules covered significantly less sensing area on the sensing layer, causing a very slow chemical reaction rate on the sensor surface. The highest responses occurred at higher NO_2_ concentrations, which was attributed to the fact that the gas molecules covered a greater area of the sensing layer. Therefore, the rate of the chemical reaction was enhanced. The response value of the sensor to 250–5000 ppb NO_2_ at 150 °C is shown in Figure 9. The response values to 250, 500, 1000, 1500, 2000, 2500, and 5000 ppb NO_2_ at 150 °C were 8.64, 9.52, 12, 16.63, 20.3, 23, and 34.22%, respectively.

Figure 10 shows the response and recovery curve of the sensor. The response time and recovery time upon exposure to 1 ppm NO_2_ was 77 s and 122 s, respectively. To investigate and verify repeatability, five reversible cycles of the gas sensor upon exposure to 1 ppm NO_2_ at 150 °C are recorded in Figure 11, which shows that the repeatability of the sensor’s response and recovery process remained close to those of the original state. The response selectivity of the sensor to different gases is shown in Figure 12. The gas sensor exhibited the highest response of 23% toward 2.5 ppm NO_2_ compared to 2.5 ppm CO, CO_2_, C_2_H_5_OH, HCHO, and NH_3_, indicating that the proposed gas sensor had excellent NO_2_ selectivity.

To investigate the effects of humidity on gas sensing, Figure 13 shows the current curve of the gas sensor with and without water molecules at 150 °C. In the initial state, there were no water molecules. Water molecules were initially introduced into the chamber. At first, the relatively humidity (RH) was 51%, and then the water molecules were removed and released from the chamber at 600 s, when the RH was 100%. The results indicated that the current decreased with increases in the relative humidity. This phenomenon was ascribed to the fact that more water vapor will cover the sensing material surface due to increases in the relatively humidity, thus competing with oxygen for the surface reaction sites, in turn leading to a decrease in the current value.

The response of the gas sensor to 1 ppm NO_2_ under different relative humidities at 150 °C is illustrated in Figure 14. The responses under 54%, 72%, 77%, 82%, and 100% RH were 8.96%, 7.3%, 6.3%, 5.8%, and 4.37%, respectively. The response value decreased with increases in the relatively humidity, which demonstrates that water molecules and oxygen molecules compete with each other. Therefore, the sensing characteristics of the gas sensor in a high humidity atmosphere declined. 

Figure 15 shows the fluctuations over a period of 30 days in the environment without an adequate package and the long-term stability of the sensor during exposure to 2.5 ppm NO_2_ at 150 °C. After one month, the response value of the sensor decreased from 23% to 5.8%. Because MZT thin film is a p-type sensing material, the concentration of the holes increases when electrons are extracted from the sensing film by oxygen (Figure 16). When NO_2_ gas molecules are introduced into the chamber, the gas molecules can capture the electrons from the sensing layer due to its higher electrophilic property. Therefore, the holes in the MZT film increase in number, leading to a decrease in resistance and an increase in the current. The interactions between NO_2_ and the chemisorbed oxygen species can be explained using the following reactions [30]:O_2(gas)_ ↔ O_2(ads)_(1)
O_2(ads)_ + 2e^−^ ↔ 2O^−^_(ads)_(2)
NO_2(gas)_ + e^−^ ↔ NO_2_^−^_(ads)_(3)
NO_2_^−^_(ads)_ + O^−^_(ads)_ + 2e^−^ ↔ NO +2O^2−^(4)

Table 1 provides a comparison between the MZT thin film gas sensor fabricated in this work and other published NO_2_ gas sensors. The obtained results are comparable with those that have been reported, demonstrating the promising potential of the MZT layer for gas sensing studies. Although there have been some research findings indicating a higher response, their sensing was achieved at gas concentrations higher than 50 ppm. In addition, few reports have mentioned NO_2_ sensing. Accordingly, we developed an MZT material that exhibits a highly selective sensing ability for low concentrations of NO_2_ gas. The good selectivity of the gas sensor in terms of NO_2_ gas can be ascribed to the high electron affinity of NO_2_ gas, the favorable sensing temperature, and the availability of electrons due to the MZT thin film in the sensor being adsorbed by the NO_2_ gas (which upon reaction with oxygen ions, causes the electrons to be released back to the sensor). This is in contrast to the behavior of NO_2_ gas, which takes more electrons from the sensing layer.

## 4. Conclusions

Sol–gel-prepared magnesium zirconia titanate thin film was used as a sensing material to sense NO_2_ performance at different operating temperatures with various concentrations of NO_2_. The incorporation of Zr in the MTO led to a rougher surface, as compared to that of MTO. A rougher surface offers more sensing area, in turn increasing the size of the adsorption sites and the gas response. The sensing properties indicated that the MZT synthesized thin film had p-type characteristics. The response of the gas sensor to 2.5 ppm NO_2_ was 2%, and the detection limit of NO_2_ dropped to 250 ppb, which was 12 times lower than that of the 3 ppm at the threshold limit value and an operating temperature of 150 °C. In addition to high stability, the proposed sensor also exhibited excellent selectivity to CO, CO_2_, C_2_H_5_OH, HCHO, and NH_3_ gases. However, humidity was found to have a significant influence on the performance of the sensor, so a good package is necessary. According to the linear fitting curves for the sensor, at a correlation coefficient between 250 and 5000 ppb and a temperature of 150 °C, the NO_2_ concentration reached 0.98. The results show potential applications for detecting quantitative NO_2_ gas molecules. As a result, the proposed simple sol–gel processed MZT thin film may have potential for NO_2_ gas sensor applications.

## Figures and Tables

**Figure 1 sensors-21-02825-f001:**
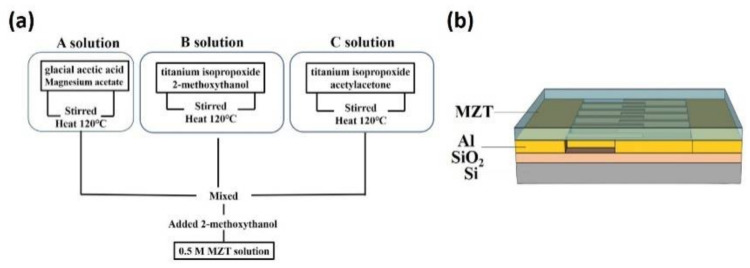
(**a**) Flow chart for preparation of 0.5 M MZT solution. (**b**) Schematic of the MZT/Al/SiO_2_/Si device and the measurement setup.

**Figure 2 sensors-21-02825-f002:**
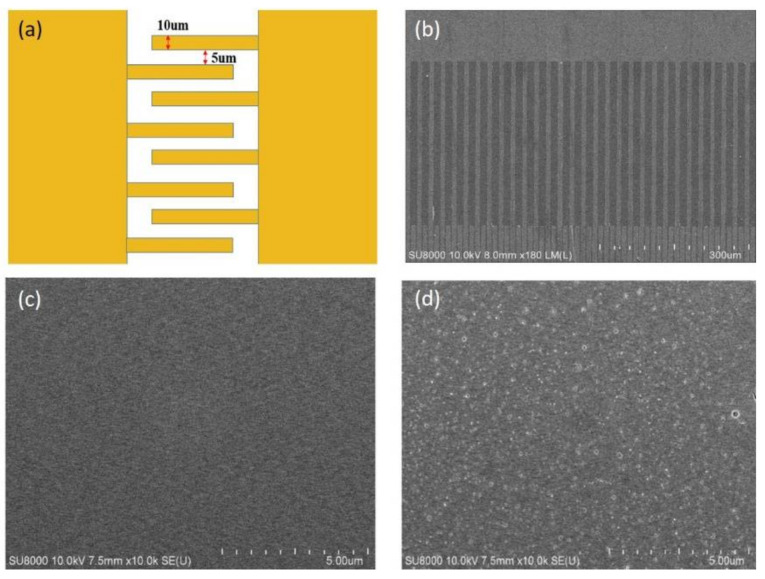
(**a**) The pattern of the interdigitated electrodes. (**b**) SEM image of the fabricated interdigitated electrodes. SEM image of the (**c**) magnesium titanate (MTO) thin film and the (**d**) magnesium zirconate titanate (MZT) thin film.

**Figure 3 sensors-21-02825-f003:**
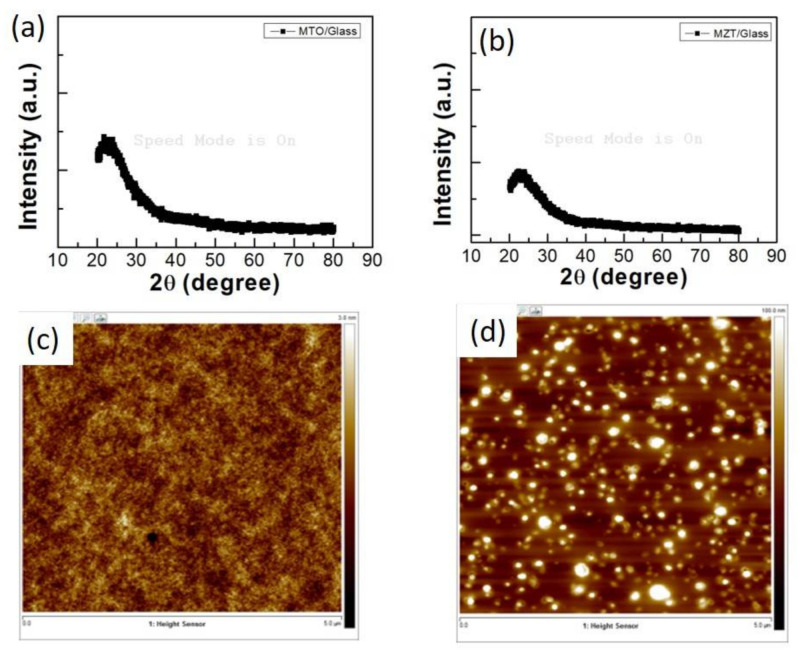
The XRD patterns of the (**a**) MTO and (**b**) MZT thin films deposited on the glass. An atomic force microscopy (AFM) image of the (**c**) MTO thin film and the (**d**) MZT thin film.

**Figure 4 sensors-21-02825-f004:**
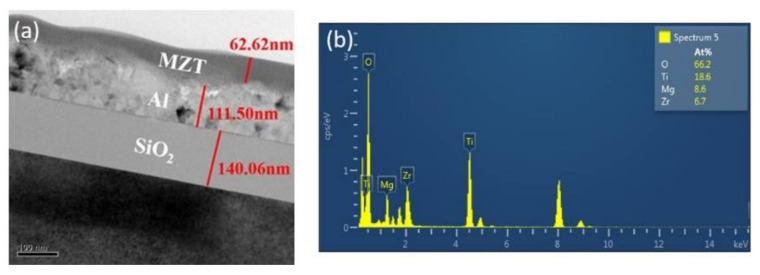
(**a**) TEM image of the MZT/Al/SiO_2_ structure. (**b**) Energy-dispersive X-ray spectroscopy (EDAX) analyses of the MZT/Al/SiO_2_/Si structure.

**Figure 5 sensors-21-02825-f005:**
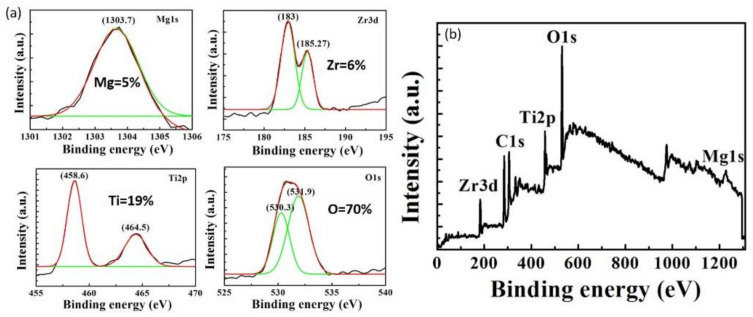
(**a**) Mg1s, Zr3d, Ti2p, and O1s peaks from the XPS spectra. (**b**) XPS survey spectrum of the MZT thin film.

**Figure 6 sensors-21-02825-f006:**
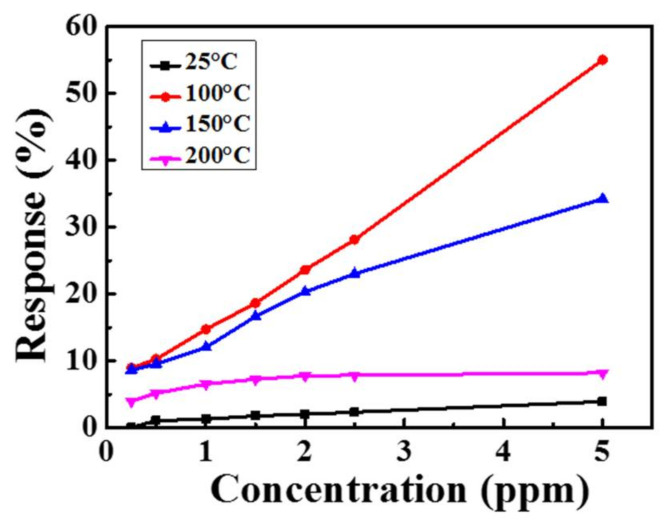
The response values of the sensor with various NO_2_ concentrations at different operating temperatures.

**Figure 7 sensors-21-02825-f007:**
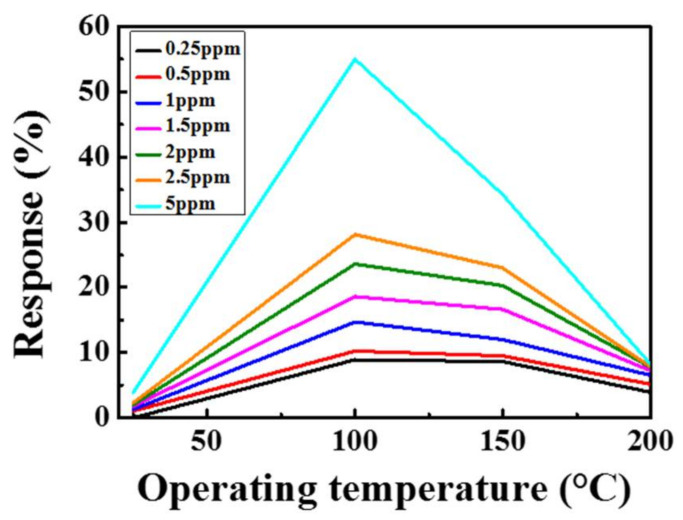
The response values of the sensor at different operating temperatures with various NO_2_ concentrations.

**Figure 8 sensors-21-02825-f008:**
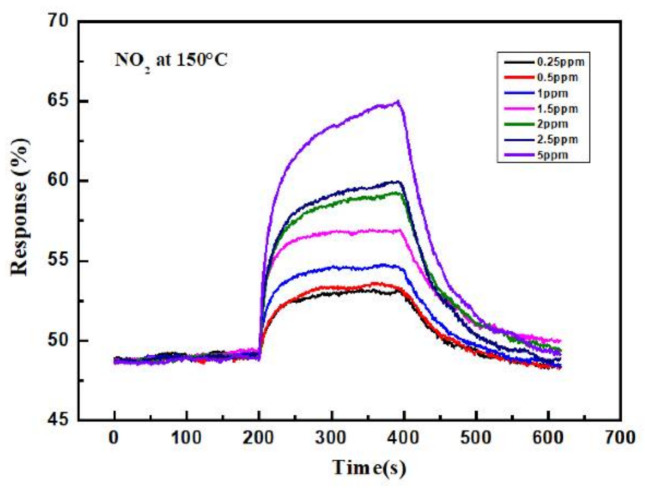
The response curve of the sensor to different concentrations of NO_2_ at 150 °C.

**Figure 9 sensors-21-02825-f009:**
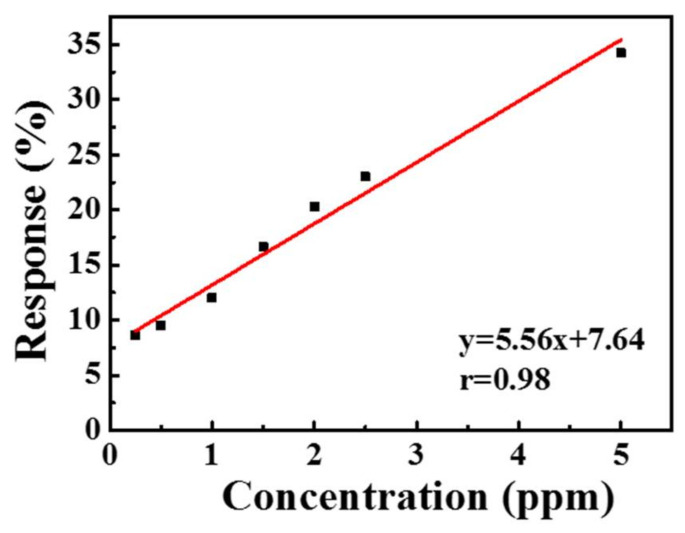
The response value of the sensor to different concentrations of NO_2_ at 150 °C.

**Figure 10 sensors-21-02825-f010:**
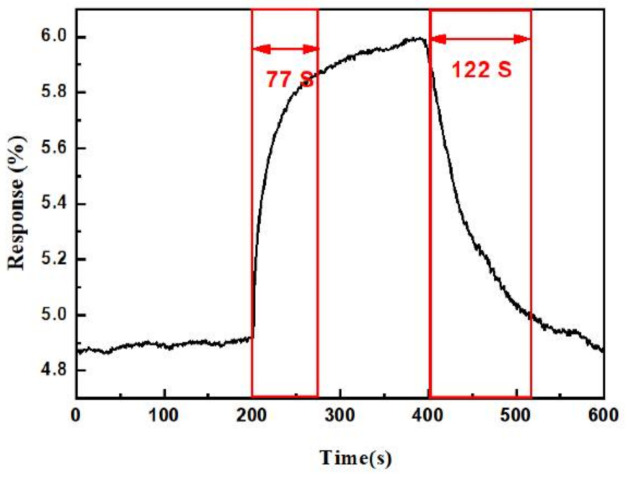
The response time and recovery time upon exposure to 1 ppm NO_2_ at 150 °C.

**Figure 11 sensors-21-02825-f011:**
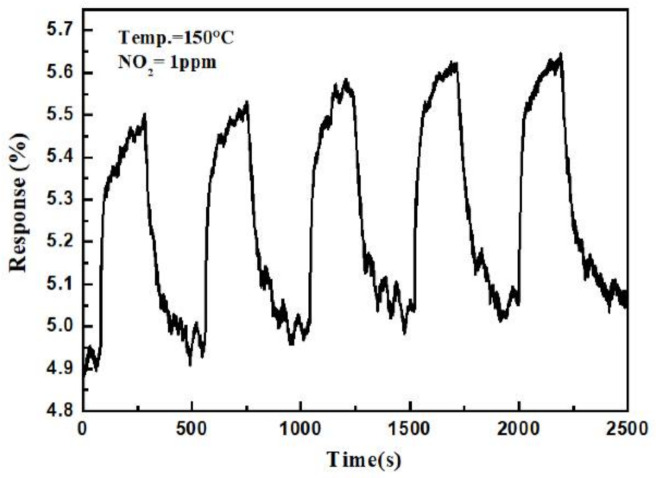
Five reversible cycles of the gas sensor to 1 ppm NO_2_ at 150 °C.

**Figure 12 sensors-21-02825-f012:**
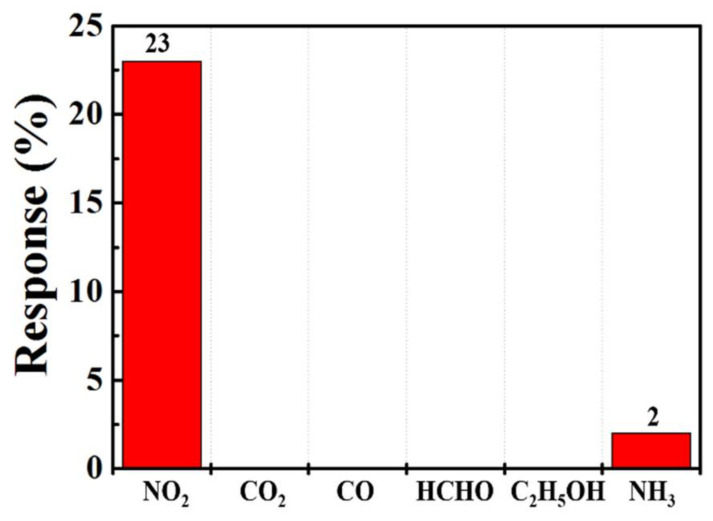
Responses of the sensor toward various target gases at 150 °C.

**Figure 13 sensors-21-02825-f013:**
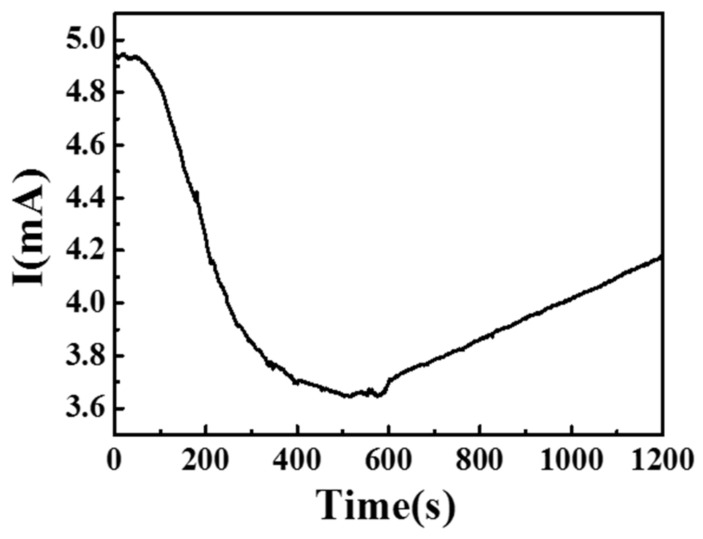
Current variations with and without water molecules at 150 °C.

**Figure 14 sensors-21-02825-f014:**
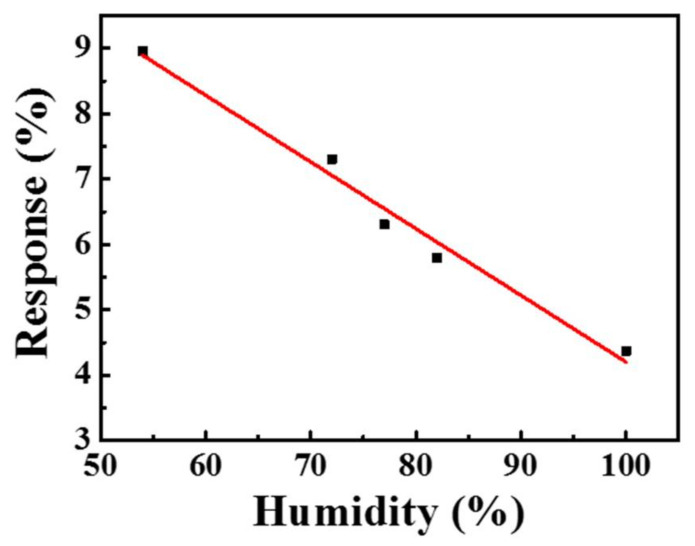
The response of the gas sensor to 1 ppm NO_2_ at 150 °C under a humidity effect.

**Figure 15 sensors-21-02825-f015:**
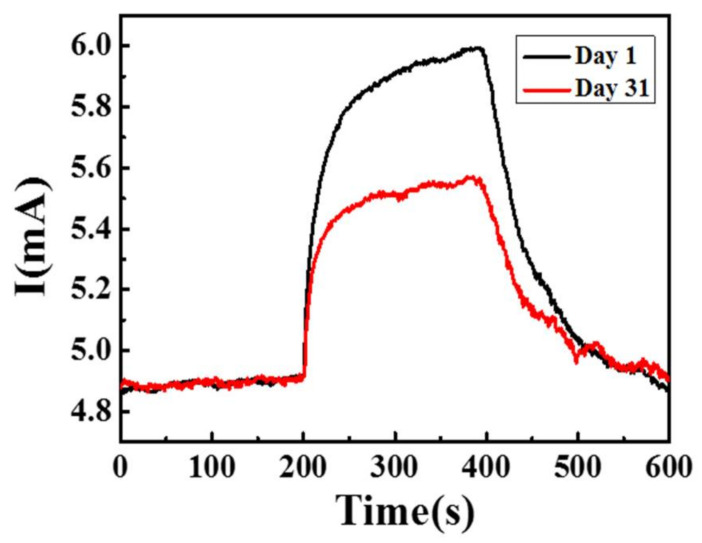
The long-term stability of the sensor to 2.5 ppm NO_2_ at 150 °C.

**Figure 16 sensors-21-02825-f016:**
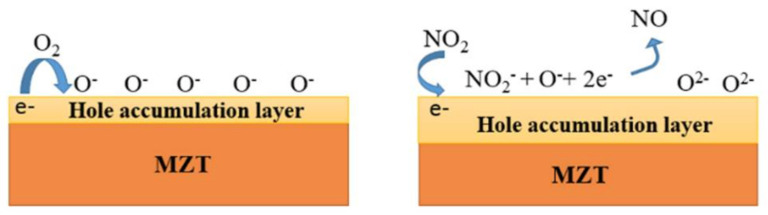
Schematic diagram of the sensing mechanism of the MZT thin film in an NO_2_ atmosphere.

**Table 1 sensors-21-02825-t001:** Comparison of the MZT thin film gas sensor with other published NO_2_ gas sensors.

Sensing Elements	Method of Preparation	Concentration (ppm)	Operating Temperature (°C)	Sensitivity (%)	Response Time (s)	Recovery Time (s)	Detection Limit (ppm)	Ref.
ZnO film	Sol-Gel	20	200	11	14	35	-	[31]
Cuo thin fiim	Thermal Evaporation	100	150	0.76	9	1200	1	[32]
TiO_2_ thin fiim	SILAR	100	250	12.78	-	-	-	[33]
2D Graphene/MoS_2_	CVD	5	150	8	-	-	0.5	[34]
CeO_2_/rGO membrane	Spray	10	RT	20.5	92	-	1	[35]
SnO/SnO_2_ thin film	RF Sputtering	10	60	4.35	165	329	1	[36]
SnO_2_ nanowire	CVD	10	RT	1	60	-	0.1	[37]
MZT thin film	Sol-Gel	2.5	150	23.0	77	122	0.25	ThisWork

## Data Availability

Not applicable.

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
