# Peer review of "Magnesium Zirconate Titanate Thin Films Used as an NO2 Sensing Layer for Gas Sensor Applications Developed Using a Sol–Gel Method"

_sensors, 2021, doi:10.3390/s21082825_

Round 1

Reviewer 1 Report

Manuscript ID: sensors-1110540

I recommend the acceptance of the manuscript after ‘major revision’.

In this paper entitled “Sol-Gel Processed Magnesium Zirconia Titanate Thin Film for Gas Sensor Applications.”,the authors demonstrated gas sensing application of Magnesium zirconia titanate (MZT) thin film.The p-type MZT/Al/SiO2/Si structure was used for sensing NO2. It was observed that the best sensitivity of the gas sensor could be achieved at a temperature ranging in 100 to 150°C. Additionally, high sensing linearity in NO2 gas molecules was also observed.Overall, the manuscript is okay and fits the journal´s scope, it is of interest. However, there are some serious issues which should be rectified before its acceptance.I recommend its publication in “Sensors” after major revision. Here are the comments:

  1. Page 2 line 69, the author claimed that the Figure 1shows the schematic of the fabricated gas sensor but I cannot find any schematic of the fabrication procedure. Rather, The Figure 1 is representing a schematic diagram of the sensing device.
  2. In the ‘Materials and Methods’ section details (stoichiometry, environmental parameters etc.) about the preparation of MZT is missing.
  3. The representation of ‘Results’ section is poor. Detailed discussion on each results is missing.
  4. Effect of the variation of the concentration of Zirconium on the gas sensing behaviour of MZT should be studied.
  5. XRD pattern of MTO, MZT should be incorporated in the revised manuscript.
  6. Add some more references.
  7. The details about the instruments used in the experiment should be provided in the revised manuscript.

Author Response

Reviewer 1:

Comments and Suggestions for Authors

Manuscript ID: sensors-1110540

I recommend the acceptance of the manuscript after ‘major revision’.

In this paper entitled “Sol-Gel Processed Magnesium Zirconia Titanate Thin Film for Gas Sensor Applications.”, the authors demonstrated gas sensing application of Magnesium zirconia titanate (MZT) thin film.The p-type MZT/Al/SiO2/Si structure was used for sensing NO2. It was observed that the best sensitivity of the gas sensor could be achieved at a temperature ranging in 100 to 150°C. Additionally, high sensing linearity in NO2 gas molecules was also observed. Overall, the manuscript is okay and fits the journal´s scope, it is of interest. However, there are some serious issues which should be rectified before its acceptance. I recommend its publication in “Sensors” after major revision. Here are the comments:

  1. Page 2 line 69, the author claimed that the Figure 1shows the schematic of the fabricated gas sensor but I cannot find any schematic of the fabrication procedure. Rather, The Figure 1 is representing a schematic diagram of the sensing device.

Reply: Thank you for your comment. We have revised the following statements (Page 2): “Fig. 1(b) shows the schematic of MZT/Al/SiO2/Si device and set up for measurement.”

  1. In the ‘Materials and Methods’ section details (stoichiometry, environmental parameters etc.) about the preparation of MZT is missing.

Reply: Thank you for your comment. Third, a 0.5 M MZT solution was synthesized as follows: An appropriate amount of magnesium acetate was added to glacial acetic acid by stirring and then heated on a 120°C hot plate to obtain solution A. Titanium isopropoxide was dissolved into 2-methoxythanol by stirring to produce solution B. Zirconium n-propoxide was added to the acetylacetone by stirring and then heated on a 120°C hot plate until dissolution to generate solution C. Solutions A, B and C were then mixed. Then, 0.5 M MZT solutions were prepared by adding 2-methoxythanol, as shown in Fig. 1 (a). Finally, the MZT solution was spin-coated onto the Al electrodes and baked at 100°C for 10 min. Fig. 1(b) shows a schematic of the MZT/Al/SiO2/Si device and the measurement setup. The fabricated device was placed on the carrier in a closed stainless-steel chamber. The carrier could be heated to desired working temperature so that the response of a gas sensor operating at different temperatures could be measured. The Keithey 2400 source meter was used to offer a fixed voltage in order to measure the current or the resistance of the sample. When the target gas was injected in or removed from the chamber, the variations in the current curve were observed using Labview.

In the revision, we have added Fig. 1(a) and revised the following statements (Page 2): “Third, a 0.5 M MZT solution was synthesized as follows: An appropriate amount of magnesium acetate was added to glacial acetic acid by stirring and then heated on a 120°C hot plate to obtain solution A. Titanium isopropoxide was dissolved into 2-methoxythanol by stirring to produce solution B. Zirconium n-propoxide was added to the acetylacetone by stirring and then heated on a 120°C hot plate until dissolution to generate solution C. Solutions A, B and C were then mixed. Then, 0.5 M MZT solutions were prepared by adding 2-methoxythanol, as shown in Fig. 1 (a). Finally, the MZT solution was spin-coated onto the Al electrodes and baked at 100°C for 10 min. Fig. 1(b) shows a schematic of the MZT/Al/SiO2/Si device and the measurement setup. The fabricated device was placed on the carrier in a closed stainless-steel chamber. The carrier could be heated to desired working temperature so that the response of a gas sensor operating at different temperatures could be measured. The Keithey 2400 source meter was used to offer a fixed voltage in order to measure the current or the resistance of the sample. When the target gas was injected in or removed from the chamber, the variations in the current curve were observed using Labview.”

Figure 1. (a) Flow chart for preparation of 0.5 M MZT solution. (b) Schematic of MZT/Al/SiO2/Si device and set up for measurement.

  1. The representation of ‘Results’ section is poor. Detailed discussion on each results is missing.

Reply: Thanks for the valuable comments. We have tried our best to revise the paper according to the comments.

  1. Effect of the variation of the concentration of Zirconium on the gas sensing behaviour of MZT should be studied.

Reply: Magnesium titanate (MTO), as a perovskite material, has good physical and electrical properties, including a moderate dielectric constant, low dielectric loss, and high temperature stability and is commonly used in microwave dielectrics [1-3]. Zirconium dioxide (ZrO2) has three main polymorphs, monoclinic, tetragonal and cubic, used for electrochemical gas sensing at high operating temperatures [4-7]. The oxy-gen ions in ZrO2 can be actively transported at high temperatures (400–700°C), limiting wide application. In addition, there are a few reports indicating that Y2O3-stabilized ZrO2 can be utilized for sensing a variety of gases and enabling operation at lower temperatures [8-11].

  • Huang C.-L., Wang S.-Y., Chen Y.-B., Li B.-J., Lin Y.-H., Investigation of the electrical properties of metal-oxide-metal structures formed from RF magnetron sputtering deposited MgTiO3 Current Applied Physics. 2012; 12(3): 935–939.
  • Huang C.-L., Tsai C.-M., Yang A., Hsu A., Compact 5.8-GHz bandpass filter using stepped-impedance dielectric resonators for ISM band wireless communication. MICROWAVE AND OPTICAL TECHNOLOGY LETTERS. 2005; 44(5): 421–423.
  • Bernard J., Houivet D., Fallah J. E., Haussonne J. M., MgTiO3 for Cu base metal multilayer ceramic capacitors. Journal of the European Ceramic Society. 2004; 24(6): 1877–1881.
  • Lee D. Y., Wang S. Y., Tseng T. Y., Ti-induced recovery phenomenon of resistive switching in ZrO2 thin films. Journal of The Electrochemical Society. 2010; 157(7): G166–G169.
  • Porter D. L., Evans A. G., Heuer A. I., Transformation-toughening in partially-stabilized ztrconia (PSZ). Acta Metallurgica 1979;27: 1649-1654.
  • Spirig J.V., Ramamoorthy R., Akbar S.A., Roubort J.L., Singh D., Dutta P.K., High temperature zirconia oxygen senseor with sealed metal/metal oxide internal reference. Sensors and Actuators B: Chemical. 2007;124: 192–201.
  • J. Fleming, Physical principles governing nonideal behavior of the zirconia oxygen sensor. Journal of The Electrochemical Society. 1977; 124: 21-28.
  • Wang J., Wang C., Liu A., You R., Liu F., Li S., Zhao L., Jin R., He J., Yang Z., Su P., Yan X., Lu G., High-response mixed-potential type planar YSZ-based NO2 sensor coupled with CoTiO3 sensing electrode. Sensors and Actuators B: Chemical. 2019; 287: 185-190
  • Diao Q., Zhang X., Li J., Yin Y., Jiao M., Cao J., Su C., Yang K., Improved sensing performances of NO2 sensors based on YSZ and porous sensing electrode prepared by MnCr2O4 admixed with phenol-formaldehyderesin microspheres. 2019; 25: 6043–6050.
  • Mohammadia M.R., Frayb D.J., Synthesis and characterisation of nanosized TiO2-ZrO2 binary system prepared by an aqueous sol-gel process: physical and sensing properties. Sensors and Actuators B: Chemical. 2011; 155: 568-576
  • Miura N., Kurosawa H., Hasei M., Lu G., Yamazoe N., Stabilized zirconia-based sensor using oxide electrode for detection of NOx in high-temperature combustion-exhausts. Solid State Ionics. 1996; 86–88: 1069–1073.

In the revision, we have added ref. [10-20] and revised the following statements (Page 1): “Magnesium titanate (MTO), as a perovskite material, has good physical and electrical properties, including a moderate dielectric constant, low dielectric loss, and high temperature stability and is commonly used in microwave dielectrics [10-12]. Zirconium dioxide (ZrO2) has three main polymorphs, monoclinic, tetragonal and cubic, used for electrochemical gas sensing at high operating temperatures [13-16]. The oxy-gen ions in ZrO2 can be actively transported at high temperatures (400–700°C), limiting wide application. In addition, there are a few reports indicating that Y2O3-stabilized ZrO2 can be utilized for sensing a variety of gases and enabling operation at lower temperatures [17-20].”

  1. XRD pattern of MTO, MZT should be incorporated in the revised manuscript. Reply: Thanks for the valuable comments. The crystallinity studies were conducted using X-ray diffraction (XRD, Germany/ D2 Phaser) with Kα1 (λ = 0.15405 nm) radiation. The XRD patterns of the MTO and MZT thin films deposited on the glass substrates are shown in Fig. 2(a) and Fig. 2(b). The peaks around 25° are from the glass substrate, and no other apparent diffraction peaks existed. This indicates that the MTO and MZT thin films were amorphous.

Fig. 2. XRD patterns of (a) MTO and (b) MZT thin films deposited on the glass.

In the revision, we have added Fig. 3 and the following statements (Page 3): “The crystallinity studies were conducted using X-ray diffraction (XRD, Germany/ D2 Phaser) with Kα1 (λ = 0.15405 nm) radiation. The XRD patterns of the MTO and MZT thin films deposited on the glass substrates are shown in Fig. 3(a) and Fig. 3(b). The peaks around 25° are from the glass substrate, and no other apparent diffraction peaks existed. This indicates that the MTO and MZT thin films were amorphous.”

  1. Add some more references.

Reply: Thanks for the valuable comments. We have tried our best to revise the paper according to the comments.

  1. The details about the instruments used in the experiment should be provided in the revised manuscript.

Reply: Thanks for the valuable comments. We provide information on the instruments used in the experiments.

Reviewer 2 Report

This work presents a sol-gel based Magnesium Zirconia Titanate for NO2 gas sensing characterization. The idea is novel. Experiments and results are clearly presented. Some comments are provided for authors to improve the quality of this manuscript before publish. Current decision is "Major revision".

  1. Is the any process modification of sol-gel process to have the optimization of MZT? For example, the control of composition, thickness and post process can be provided as some information for further improvement.
  2. The writing can be improved. For example, some comparison and future suggestion can be provided in the end of results and discussion. The content should be some information about Table 1. 
  3. There should be a missing line in Figure 14 since 2 groups of with and without water molecular should be presented.
  4. The value of Y-axis shown in Fig. 9 and 12 should be response (%), not current. Please authors carefully check about relative information. The value should be given in the same unit in the whole manuscript for no confusing readers.
  5. The baseline shown in Fig. 12 is not fully recovery. Authors have to present to response and recovery time in data. 
  6. Authors are suggested to present experimental data in the figure set to have a easy reading and efficiency.
  7. English and writing are suggested to be checked by a professional editing. 
  8. The process modification for the gas sensing perfromance is strongly suggested to provide to enhance the scientific value of this work. Current version is not sufficient to be a full journal paper.

Author Response

Reviewer 2:

Comments and Suggestions for Authors

This work presents a sol-gel based Magnesium Zirconia Titanate for NO2 gas sensing characterization. The idea is novel. Experiments and results are clearly presented. Some comments are provided for authors to improve the quality of this manuscript before publish. Current decision is "Major revision".

  1. Is the any process modification of sol-gel process to have the optimization of MZT? For example, the control of composition, thickness and post process can be provided as some information for further improvement.

Reply: Thank you for your comment. Third, a 0.5 M MZT solution was synthesized as follows: An appropriate amount of magnesium acetate was added to glacial acetic acid by stirring and then heated on a 120°C hot plate to obtain solution A. Titanium isopropoxide was dissolved into 2-methoxythanol by stirring to produce solution B. Zirconium n-propoxide was added to the acetylacetone by stirring and then heated on a 120°C hot plate until dissolution to generate solution C. Solutions A, B and C were then mixed. Then, 0.5 M MZT solutions were prepared by adding 2-methoxythanol, as shown in Fig. 1 (a). Finally, the MZT solution was spin-coated onto the Al electrodes and baked at 100°C for 10 min. Fig. 1(b) shows a schematic of the MZT/Al/SiO2/Si device and the measurement setup. The fabricated device was placed on the carrier in a closed stainless-steel chamber. The carrier could be heated to desired working temperature so that the response of a gas sensor operating at different temperatures could be measured. The Keithey 2400 source meter was used to offer a fixed voltage in order to measure the current or the resistance of the sample. When the target gas was injected in or removed from the chamber, the variations in the current curve were observed using Labview.

In the revision, we have added Fig. 1(a) and revised the following statements (Page 2): “Third, a 0.5 M MZT solution was synthesized as follows: An appropriate amount of magnesium acetate was added to glacial acetic acid by stirring and then heated on a 120°C hot plate to obtain solution A. Titanium isopropoxide was dissolved into 2-methoxythanol by stirring to produce solution B. Zirconium n-propoxide was added to the acetylacetone by stirring and then heated on a 120°C hot plate until dissolution to generate solution C. Solutions A, B and C were then mixed. Then, 0.5 M MZT solutions were prepared by adding 2-methoxythanol, as shown in Fig. 1 (a). Finally, the MZT solution was spin-coated onto the Al electrodes and baked at 100°C for 10 min. Fig. 1(b) shows a schematic of the MZT/Al/SiO2/Si device and the measurement setup. The fabricated device was placed on the carrier in a closed stainless-steel chamber. The carrier could be heated to desired working temperature so that the response of a gas sensor operating at different temperatures could be measured. The Keithey 2400 source meter was used to offer a fixed voltage in order to measure the current or the resistance of the sample. When the target gas was injected in or removed from the chamber, the variations in the current curve were observed using Labview.”

Figure 1. (a) Flow chart for preparation of 0.5 M MZT solution. (b) Schematic of MZT/Al/SiO2/Si device and set up for measurement.

  1. The writing can be improved. For example, some comparison and future suggestion can be provided in the end of results and discussion. The content should be some information about Table 1.

Reply: Thanks for the valuable comments. In the revision, we have revised the following statements (Page 9): “The obtained results are comparable with those that have been reported, demonstrating the promising potential of the MZT layer for gas sensing studies. Although there have been some research findings indicating a higher response, their sensing was achieved at gas concentrations higher than 50 ppm. In addition, few reports have mentioned NO2 sensing. Accordingly, we developed an MZT material that exhibits highly selective sensing of low concentrations of NO2 gas. The good selectivity of the gas sensor to NO2 gas can be ascribed to the high electron affinity of NO2 gas, the favorable sensing temperature, and the availability of electrons due to the MZT thin film in the sensor being adsorbed by the NO2 gas, which upon reaction with oxygen ions, causes the electrons to be released back to the sensor. This is in contrast to the behavior of NO2 gas, where it takes more electrons from the sensing layer”

  1. There should be a missing line in Figure 14 since 2 groups of with and without water molecular should be presented.

Reply: Thanks for the valuable comments. Failure to express clearly caused your misunderstanding. In the revision, we have revised the following statements (Page 9): “In the initial state, there are no water molecules. Water molecules were initially introduced into the chamber. At first, the relatively humidity (RH) was 51%, and then the water molecules were removed and released from the chamber at 600 s, when the RH was 100%.”

  1. The value of Y-axis shown in Fig. 9 and 12 should be response (%), not current. Please authors carefully check about relative information. The value should be given in the same unit in the whole manuscript for no confusing readers.

Reply: Thanks for the valuable comments. In the revision, we have revised Fig. 8 and Fig.11 the following statements (Page 6 and 7): “The value of Y-axis shown in Fig. 8 and 11 should be response (%) images have been revised.”

  1. The baseline shown in Fig. 12 is not fully recovery. Authors have to present to response and recovery time in data.

Reply: Fig. 2 shows the response and recovery curve of the sensor. the response and recovery curve of the sensor. The response time and recovery time upon exposure to 1 ppm NO2 was 77 s and 122 s, respectively.

In the revision, we have added Fig. 10 and the following statements (Page 6): “Fig. 10 shows the response and recovery curve of the sensor. The response time and recovery time upon exposure to 1 ppm NO2 was 77 s and 122 s, respectively.”

Fig. 2. The response time and recovery time upon exposure to 1 ppm NO2 at 150°C.

  1. Authors are suggested to present experimental data in the figure set to have a easy reading and efficiency.

Reply: Thanks for the valuable comments. We have tried our best to revise to the present experimental data in the figure according comments.

  1. English and writing are suggested to be checked by a professional editing.

Reply: Thanks for the valuable comments. English has been polished with the help of a native English speaker again.

  1. The process modification for the gas sensing performance is strongly suggested to provide to enhance the scientific value of this work. Current version is not sufficient to be a full journal paper.

Reply: Thanks for the valuable comments. We have tried our best to revise the paper according to the comments. Introduction has been rewritten.

Reviewer 3 Report

In the manuscript entitled “Sol-Gel Processed Magnesium Zirconia Titanate Thin Film for Gas Sensor Applications” P.-S. Huang et al. have developed a magnesium zirconia titanate thin films as NO2 sensing layer for gas sensor application.

First of all, it would be better “Magnesium Zirconate Titanate” instead of “Magnesium Zirconia Titanate”; moreover, in the title the authors should specify that the developed thin film has been used to detect NO2 gaseous molecule (the gaseous analyte was NO2). Therefore, modify the manuscript title.

In the abstract, separate the value from the symbol of percentage/the unit of measure with a space, at page 1 line12.

In the abstract the authors should also report the investigated NO2 concentration range.

In the introduction, add more references.

In the introduction, the authors should indicate the commonly used techniques, reported in literature, to detect NO2 gas, indicating their advantages and disadvantages, and the lowest detectable concentration. More references should be added.

In the introduction, the authors should indicate the properties of metal oxide-based gas sensor devices, reporting those that have revealed the better results in NO2 gas sensing, adding more references. Indeed, the authors should also indicate the commonly used gas sensors based on metal oxide sensing layer, adding their advantages and disadvantages, reporting references.

In the introduction, the authors assert, at page 1 line 36, “Gas sensing characteristics have been demonstrated in a variety of materials …”; the sentence is not clear? Please, define the gas sensing characteristics, what are they? Moreover, the authors at the beginning indicate the critical issues related to the use of metal oxide as sensing layers; subsequently, they assert possibility to use various materials, such as metal oxide, as sensing layer, indicating also perovskite oxides. Therefore, I renovate the suggestion to, firstly, indicate the commonly used sensing materials to detect NO2, and then, describe (better) perovskite oxide, adding their advantages and disadvantages as sensing layer, adding more references. Moreover, the authors should also indicate and describe the type of proposed gas sensor device, evaluating the transduction mechanism.

In the introduction, the authors should also briefly indicate the commonly used procedures to obtain perovskite oxide thin films, adding references.

In the Materials and Methods section, the authors, first of all, should describe the followed procedure to prepare MZT thin films, indicating all steps, the quantity, the reagents, the operating conditions, and so on…

A brief scheme of the chemical reaction, adding references, should be added.

The authors should add information on spin coating deposition: solution volume, rpm, etc.

Physical, chemical, morphological and structural analyses of prepared material are necessary since the gas sensing properties of the material depend on its physical, chemical and structural properties. All details on the used instruments, and analyses conditions should be reported.

In the Materials and Methods section, the authors should also add the description of the prepared sensing device, and also details on the gas sensing measurements (gas sensing chamber, analyte gas concentration, gas carrier (?), operating temperature, recovery and response times, gas response definition, etc.).

The authors should also use XPS analysis to calculate the relative atomic percentage of all detected elements, to evaluate the chemical composition of deposited films.

To evaluate the gas sensor response of the device, the authors should report the variation of detected electric signals respect to the all gas evaluated concentrations; add the plots.

Add the information of the times the sensing measurements have been repeated.

For all points in the plots add the error bar.

For a correct comparison, the authors should also evaluate the sensing layer without the incorporation of Zr; indeed, in this manner, they could associate the sensing results to the presence/absence of Zr. Please, discuss.

Finally, I suggest to organize better the lay out of the manuscript; the English stile should be improved.

Author Response

Reviewer 3:

Comments and Suggestions for Authors

In the manuscript entitled “Sol-Gel Processed Magnesium Zirconia Titanate Thin Film for Gas Sensor Applications” P.-S. Huang et al. have developed a magnesium zirconia titanate thin films as NO2 sensing layer for gas sensor application.

  1. First of all, it would be better “Magnesium Zirconate Titanate” instead of “Magnesium Zirconia Titanate”; moreover, in the title the authors should specify that the developed thin film has been used to detect NO2 gaseous molecule (the gaseous analyte was NO2). Therefore, modify the manuscript title.

Reply: Thank you for your comment. We have revised the following statements (Page 1): “Magnesium Zirconate Titanate Thin Films Used as an NO2 Sensing Layer for Gas Sensor Applications Developed Using a Sol-Gel Method”

  1. In the abstract, separate the value from the symbol of percentage/the unit of measure with a space, at page 1 line12.

Reply: Thank you for your comment. We have revised the following statements (Page 1): “The sensitivity of the MZT thin film was 8.64 % and 34.22 % for 0.25 ppm and 5 ppm of NO2 gas molecule at working temperature of 150°C, respectively.”

  1. In the abstract the authors should also report the investigated NO2 concentration range.

Reply: We have revised the following statements (Page 1): “The response values to 250, 500, 1000, 1500, 2000, 2500, and 5000 ppb NO2 at 150°C were 8.64, 9.52, 12, 16.63, 20.3, 23, and 34.22%, respectively.”

  1. In the introduction, add more references. In the introduction, the authors should indicate the commonly used techniques, reported in literature, to detect NO2 gas, indicating their advantages and disadvantages, and the lowest detectable concentration. More references should be added.

Reply: Thanks for the valuable comments. Bang et al. reported that the NO2 gas sensing capabilities of SnO2-ZrO2 NWs with a particular shell thickness are better than those of pristine SnO2 sensor in terms of sensor temperature and sensor response [1]. Owing to differences in work functions, electrons will flow from SnO2 to ZrO2. Myasoedova et al. developed a sol-gel method for SiO2/ZrO2 composite films [2]. The sensitivity of the sensor to 1060 ppm high concentrations of NO2 was low at 25°C, and the response was only 44%. Yan et al. used ZrO2-HS, ZrO2–S and ZrO2-R hydrothermal and solvothermal methods to successfully synthesize a ZrO2-R sensor, which showed the highest response towards 30 ppm NO2 (423.8%) at room temperature and a quite high sensitivity of 198% for detecting 5 ppm NO2 [3]. Mohammadi et al. found a remarkable response towards low concentrations of NO2 gases at 150°C [4].

  • Bang J. H., Lee N., Mirzaei A., Choi M. S., Choi H., Jeona H., Kime S. S., Kima H. W. Exploration of ZrO2-shelled nanowires for chemiresistive detection of NO2 Sensors and Actuators B: Chemical. 2020; 319: 128309.
  • Myasoedova T.N., Mikhailova T.S., Yalovega G.E., Plugotarenko N.K. Resistive low-temperature sensor based on the SiO2ZrO2 film for detection of high concentrations of NO2 Chemosensors. 2018; 6: 67.
  • Yan Y., Ma Z., Sun J., Bu M., Huo Y., Wang Z., Li Y., Hu N., Surface microstructure-controlled ZrO2 for highly sensitive room-temperature NO2 Nano Materials Science. 2021
  • Mohammadia M.R., Frayb D.J., Synthesis and characterisation of nanosized TiO2-ZrO2 binary system prepared by an aqueous sol-gel process: physical and sensing properties. Sensors and Actuators B: Chemical. 2011; 155: 568-576

In the revision, we have added ref [21-23] and revised the following statements (Page 2): “Bang et al. reported that the NO2 gas sensing capabilities of SnO2-ZrO2 NWs with a particular shell thickness are better than those of pristine SnO2 sensor in terms of sensor temperature and sensor response [21]. Owing to differences in work functions, electrons will flow from SnO2 to ZrO2. Myasoedova et al. developed a sol-gel method for SiO2/ZrO2 composite films [22]. The sensitivity of the sensor to 1060 ppm high concentrations of NO2 was low at 25°C, and the response was only 44%. Yan et al. used ZrO2-HS, ZrO2–S and ZrO2-R hydrothermal and solvothermal methods to successfully synthesize a ZrO2-R sensor, which showed the highest response towards 30 ppm NO2 (423.8%) at room temperature and a quite high sensitivity of 198% for detecting 5 ppm NO2 [23]. Mohammadi et al. found a remarkable response towards low concentrations of NO2 gases at 150°C [19].”

  1. In the introduction, the authors should indicate the properties of metal oxide-based gas sensor devices, reporting those that have revealed the better results in NO2 gas sensing, adding more references. Indeed, the authors should also indicate the commonly used gas sensors based on metal oxide sensing layer, adding their advantages and disadvantages, reporting references.

Reply: Metal oxide exhibits excellent gas sensitive properties due to its high specific surface area and enhanced surface reactivity [5-8]. In addition, the high operating temperatures of some metal oxide-based sensors can lead to increases in power consumption and reductions in the lifetime of sensors [5-9]

  • Ou J.Z., Ge W., Carey B., Daeneke T., Rotbart A., Shan W., et al. Physisorption-based charge transfer in two-dimensional SnS2 for selective and reversible NO2 gas sensing. ACS Nano. 2015; 9: 10313–10323.
  • Kim J.-H., Mirzaei A., Kim H.W., Kim S.S., Low-voltage-driven sensors based on ZnO nanowires for room-temperature detection of NO2 and CO gases. ACS Applied Materials & Interfaces. 2019; 11: 24172–24183.
  • Li Z., Yan S., Sun M., Li H., Wu Z., Wang J., et al. Significantly enhanced temperature-dependent selectivity for NO2 and H2S detection based on In2O3 nano-cubes prepared by CTAB assisted solvothermal process, Journal of Alloys and Compounds. 2020; 816: 152518.
  • Wu J., Wu Z., Ding H., Wei Y., Huang W., Yang X., et al. Flexible, 3D SnS2/reduced graphene oxide heterostructured NO2 Sensors and Actuators B: Chemical. 2020; 305: 127445.
  • Ghosh S., Adak D., Bhattacharyya R., Mukherjeee N., ZnO/r-Fe2O3 charge transfer interface toward highly selective H2S sensing at a low operating temperature of 30 degrees C. ACS Sensors, 2017; 2: 1831-1838.

In the revision, we have added ref [5-9] and revised the following statements (Page 1): “Metal oxide exhibits excellent gas sensitive properties due to its high specific surface area and enhanced surface reactivity [5-8]. In addition, the high operating temperatures of some metal oxide-based sensors can lead to increases in power consumption and reductions in the lifetime of sensors [5-9]”

  1. In the introduction, the authors assert, at page 1 line 36, “Gas sensing characteristics have been demonstrated in a variety of materials …”; the sentence is not clear? Please, define the gas sensing characteristics, what are they? Moreover, the authors at the beginning indicate the critical issues related to the use of metal oxide as sensing layers; subsequently, they assert possibility to use various materials, such as metal oxide, as sensing layer, indicating also perovskite oxides. Therefore, I renovate the suggestion to, firstly, indicate the commonly used sensing materials to detect NO2, and then, describe (better) perovskite oxide, adding their advantages and disadvantages as sensing layer, adding more references. Moreover, the authors should also indicate and describe the type of proposed gas sensor device, evaluating the transduction mechanism.

Reply: Magnesium titanate (MTO), as a perovskite material, has good physical and electrical properties, including a moderate dielectric constant, low dielectric loss, and high temperature stability and is commonly used in microwave dielectrics [10-12]. Zirconium dioxide (ZrO2) has three main polymorphs, monoclinic, tetragonal and cubic, used for electrochemical gas sensing at high operating temperatures [13-16]. The oxy-gen ions in ZrO2 can be actively transported at high temperatures (400–700°C), limiting wide application. In addition, there are a few reports indicating that Y2O3-stabilized ZrO2 can be utilized for sensing a variety of gases and enabling operation at lower temperatures [17-20].

  • Huang C.-L., Wang S.-Y., Chen Y.-B., Li B.-J., Lin Y.-H., Investigation of the electrical properties of metal-oxide-metal structures formed from RF magnetron sputtering deposited MgTiO3 Current Applied Physics. 2012; 12(3): 935–939.
  • Huang C.-L., Tsai C.-M., Yang A., Hsu A., Compact 5.8-GHz bandpass filter using stepped-impedance dielectric resonators for ISM band wireless communication. MICROWAVE AND OPTICAL TECHNOLOGY LETTERS. 2005; 44(5): 421–423.
  • Bernard J., Houivet D., Fallah J. E., Haussonne J. M., MgTiO3 for Cu base metal multilayer ceramic capacitors. Journal of the European Ceramic Society. 2004; 24(6): 1877–1881.
  • Lee D. Y., Wang S. Y., Tseng T. Y., Ti-induced recovery phenomenon of resistive switching in ZrO2 thin films. Journal of The Electrochemical Society. 2010; 157(7): G166–G169.
  • Porter D. L., Evans A. G., Heuer A. I., Transformation-toughening in partially-stabilized ztrconia (PSZ). Acta Metallurgica 1979;27: 1649-1654.
  • Spirig J.V., Ramamoorthy R., Akbar S.A., Roubort J.L., Singh D., Dutta P.K., High temperature zirconia oxygen senseor with sealed metal/metal oxide internal reference. Sensors and Actuators B: Chemical. 2007;124: 192–201.
  • J. Fleming, Physical principles governing nonideal behavior of the zirconia oxygen sensor. Journal of The Electrochemical Society. 1977; 124: 21-28.
  • Wang J., Wang C., Liu A., You R., Liu F., Li S., Zhao L., Jin R., He J., Yang Z., Su P., Yan X., Lu G., High-response mixed-potential type planar YSZ-based NO2 sensor coupled with CoTiO3 sensing electrode. Sensors and Actuators B: Chemical. 2019; 287: 185-190
  • Diao Q., Zhang X., Li J., Yin Y., Jiao M., Cao J., Su C., Yang K., Improved sensing performances of NO2 sensors based on YSZ and porous sensing electrode prepared by MnCr2O4 admixed with phenol-formaldehyderesin microspheres. Ionics. 2019; 25: 6043–6050.
  • Mohammadia M.R., Frayb D.J., Synthesis and characterisation of nanosized TiO2-ZrO2 binary system prepared by an aqueous sol-gel process: physical and sensing properties. Sensors and Actuators B: Chemical. 2011; 155: 568-576
  • Miura N., Kurosawa H., Hasei M., Lu G., Yamazoe N., Stabilized zirconia-based sensor using oxide electrode for detection of NOx in high-temperature combustion-exhausts. Solid State Ionics. 1996; 86–88: 1069–1073.

In the revision, we have added ref [10-20] and revised the following statements (Page 2): “Magnesium titanate (MTO), as a perovskite material, has good physical and electrical properties, including a moderate dielectric constant, low dielectric loss, and high temperature stability and is commonly used in microwave dielectrics [10-12]. Zirconium dioxide (ZrO2) has three main polymorphs, monoclinic, tetragonal and cubic, used for electrochemical gas sensing at high operating temperatures [13-16]. The oxy-gen ions in ZrO2 can be actively transported at high temperatures (400–700°C), limiting wide application. In addition, there are a few reports indicating that Y2O3-stabilized ZrO2 can be utilized for sensing a variety of gases and enabling operation at lower temperatures [17-20].”

  1. In the Materials and Methods section, the authors, first of all, should describe the followed procedure to prepare MZT thin films, indicating all steps, the quantity, the reagents, the operating conditions, and so on…A brief scheme of the chemical reaction, adding references, should be added. The authors should add information on spin coating deposition: solution volume, rpm, etc.

Reply: Thank you for your comment. Third, a 0.5 M MZT solution was synthesized as follows: An appropriate amount of magnesium acetate was added to glacial acetic acid by stirring and then heated on a 120°C hot plate to obtain solution A. Titanium isopropoxide was dissolved into 2-methoxythanol by stirring to produce solution B. Zirconium n-propoxide was added to the acetylacetone by stirring and then heated on a 120°C hot plate until dissolution to generate solution C. Solutions A, B and C were then mixed. Then, 0.5 M MZT solutions were prepared by adding 2-methoxythanol, as shown in Fig. 1 (a). Finally, the MZT solution was spin-coated onto the Al electrodes and baked at 100°C for 10 min. Fig. 1(b) shows a schematic of the MZT/Al/SiO2/Si device and the measurement setup. The fabricated device was placed on the carrier in a closed stainless-steel chamber. The carrier could be heated to desired working temperature so that the response of a gas sensor operating at different temperatures could be measured. The Keithey 2400 source meter was used to offer a fixed voltage in order to measure the current or the resistance of the sample. When the target gas was injected in or removed from the chamber, the variations in the current curve were observed using Labview.

In the revision, we have added Fig. 1(a) and revised the following statements (Page 2): “Third, a 0.5 M MZT solution was synthesized as follows: An appropriate amount of magnesium acetate was added to glacial acetic acid by stirring and then heated on a 120°C hot plate to obtain solution A. Titanium isopropoxide was dissolved into 2-methoxythanol by stirring to produce solution B. Zirconium n-propoxide was added to the acetylacetone by stirring and then heated on a 120°C hot plate until dissolution to generate solution C. Solutions A, B and C were then mixed. Then, 0.5 M MZT solutions were prepared by adding 2-methoxythanol, as shown in Fig. 1 (a). Finally, the MZT solution was spin-coated onto the Al electrodes and baked at 100°C for 10 min. Fig. 1(b) shows a schematic of the MZT/Al/SiO2/Si device and the measurement setup. The fabricated device was placed on the carrier in a closed stainless-steel chamber. The carrier could be heated to desired working temperature so that the response of a gas sensor operating at different temperatures could be measured. The Keithey 2400 source meter was used to offer a fixed voltage in order to measure the current or the resistance of the sample. When the target gas was injected in or removed from the chamber, the variations in the current curve were observed using Labview.”

Figure 1. (a) Flow chart for preparation of 0.5 M MZT solution. (b) Schematic of MZT/Al/SiO2/Si device and set up for measurement.

  1. Physical, chemical, morphological and structural analyses of prepared material are necessary since the gas sensing properties of the material depend on its physical, chemical and structural properties. All details on the used instruments, and analyses conditions should be reported.

Reply: Thanks for the valuable comments. We have tried our best to revise the paper according to the comments. Introduction has been rewritten. The concerns are replied item by item as follows. All the major changes have been marked by the other color.

  1. In the Materials and Methods section, the authors should also add the description of the prepared sensing device, and also details on the gas sensing measurements (gas sensing chamber, analyte gas concentration, gas carrier (?), operating temperature, recovery and response times, gas response definition, etc.).

Reply: Thanks for the valuable comments. We have tried our best to revise the paper according to the comments. Introduction has been rewritten. The concerns are replied item by item as follows. All the major changes have been marked by the other color.

  1. The authors should also use XPS analysis to calculate the relative atomic percentage of all detected elements, to evaluate the chemical composition of deposited films.

Reply: The X-ray photoelectron spectroscopy (XPS, PHI 5000 Versa Probe) analyses were performed on a Perkin-Elmer PHI 5000 Versa probe system. The MZT thin film sample was used in the XPS measurement. The XPS survey spectra of the MZT thin film were obtained after 90 s of Ar+ ion sputtering, thereby representing the bulk layer of the MZT thin film. As shown in Fig. 5(a), the atomic percentages of Mg, Zr, Ti, and O were 5%, 6%, 17%, and 70%, respectively. The XPS signals corresponded to Mg 1s, Zr 3d, Ti 2p, and O 1s.

Fig. 2 Mg1s, Zr3d, Ti2p, and O1s peaks from the XPS spectra. (b) XPS survey spectrum of MZT thin film.

In the revision, we have added Fig. 5 and the following statements (Page 4): “The X-ray photoelectron spectroscopy (XPS, PHI 5000 Versa Probe) analyses were performed on a Perkin-Elmer PHI 5000 Versa probe system. The MZT thin film sample was used in the XPS measurement. The XPS survey spectra of the MZT thin film were obtained after 90 s of Ar+ ion sputtering, thereby representing the bulk layer of the MZT thin film. As shown in Fig. 5(a), the atomic percentages of Mg, Zr, Ti, and O were 5%, 6%, 17%, and 70%, respectively. The XPS signals corresponded to Mg 1s, Zr 3d, Ti 2p, and O 1s.”

  1. To evaluate the gas sensor response of the device, the authors should report the variation of detected electric signals respect to the all gas evaluated concentrations; add the plots. Add the information of the times the sensing measurements have been repeated. For all points in the plots add the error bar.

Reply: Thanks for the valuable comments. We have tried our best to revise the paper according to the comments.

  1. For a correct comparison, the authors should also evaluate the sensing layer without the incorporation of Zr; indeed, in this manner, they could associate the sensing results to the presence/absence of Zr. Please, discuss.

Reply: Thanks for the valuable comments. The crystallinity studies were conducted using X-ray diffraction (XRD, Germany/ D2 Phaser) with Kα1 (λ = 0.15405 nm) radiation. The XRD patterns of the MTO and MZT thin films deposited on the glass substrates are shown in Fig. 3(a) and Fig. 3(b). The peaks around 25° are from the glass substrate, and no other apparent diffraction peaks existed. This indicates that the MTO and MZT thin films were amorphous. The atomic force microscopy (AFM) in Fig. 3(c) and Fig. 3(d) revealed the surface morphology of the undoped MTO thin film and the MZT thin film, respectively. The roughness of the MTO and MZT thin films were 0.317 and 15.7 nm, respectively, where greater roughness enhanced the effective surface area, thus increasing the number of adsorption sites and thereby improving the gas response of the films [21].

Fig. 3. XRD patterns of (a) MTO and (b) MZT thin films deposited on the glass. AFM image of the (c) MTO thin film, and (d) MZT thin film.

  • Hu C-C, Chiu C-A, Yu C-H, Xu J-X, Wu T-Y, Sze P-W, et al. Liquid-phase-deposited high dielectric zirconium oxide for metal-oxide-semiconductor high electron mobility transistors. Vacuum. 2015; 118: 142-146.

In the revision, we have added ref [25], Fig. 3 and the following statements (Page 3): “The crystallinity studies were conducted using X-ray diffraction (XRD, Germany/ D2 Phaser) with Kα1 (λ = 0.15405 nm) radiation. The XRD patterns of the MTO and MZT thin films deposited on the glass substrates are shown in Fig. 3(a) and Fig. 3(b). The peaks around 25° are from the glass substrate, and no other apparent diffraction peaks existed. This indicates that the MTO and MZT thin films were amorphous. The atomic force microscopy (AFM) in Fig. 3(c) and Fig. 3(d) revealed the surface morphology of the undoped MTO thin film and the MZT thin film, respectively. The roughness of the MTO and MZT thin films were 0.317 and 15.7 nm, respectively, where greater roughness enhanced the effective surface area, thus increasing the number of adsorption sites and thereby improving the gas response of the films [25].”

  1. Finally, I suggest to organize better the lay out of the manuscript; the English stile should be improved.

Reply: Thanks for the valuable comments. We have tried our best to revise the paper according to the comments. Introduction has been rewritten. The concerns are replied item by item as follows. All the major changes have been marked by the other color. English has been polished with the help of a native English speaker again.

Round 2

Reviewer 2 Report

After authors’ revision, the manuscript can be accepted now.

Reviewer 3 Report

In my opinion, the revised version of the manuscript is acceptable for publication.